# Endoglin Targeting: Lessons Learned and Questions That Remain

**DOI:** 10.3390/ijms22010147

**Published:** 2020-12-25

**Authors:** Yingmiao Liu, Madelon Paauwe, Andrew B. Nixon, Lukas J.A.C. Hawinkels

**Affiliations:** 1Department of Medicine, Duke University Medical Center, Durham, NC 27710, USA; yl44@duke.edu (Y.L.); anixon@duke.edu (A.B.N.); 2Department of Gastroenterology and Hepatology, Leiden University Medical Center, 2300 RC Leiden, The Netherlands; madelon.paauwe@gmail.com

**Keywords:** endoglin, TRC105, angiogenesis, clinical trials, biomarkers

## Abstract

Approximately 30 years ago, endoglin was identified as a transforming growth factor (TGF)-β coreceptor with a crucial role in developmental biology and tumor angiogenesis. Its selectively high expression on tumor vessels and its correlation with poor survival in cancer patients led to the exploration of endoglin as a therapeutic target for cancer. The endoglin neutralizing antibody TRC105 (Carotuximab^®^, Tracon Pharmaceuticals (San Diego, CA, USA) was subsequently tested in a wide variety of preclinical cancer models before being tested in phase I-III clinical studies in cancer patients as both a monotherapy and in combination with other chemotherapeutic and anti-angiogenic therapies. The combined data of these studies have revealed new insights into the role of endoglin in angiogenesis and its expression and functional role on other cells in the tumor microenvironment. In this review, we will summarize the preclinical work, clinical trials and biomarker studies of TRC105 and explore what these studies have enabled us to learn and what questions remain unanswered.

## 1. Introduction

### 1.1. Introduction and Endoglin History

Endoglin is a 180 kDa RGD-domain-containing transmembrane glycoprotein, which was first identified on endothelial cells in 1990, whose expression is increased by hypoxia [1]. Later, its role as a part of the transforming growth factor (TGF)-β receptor signaling complex [2] was shown, followed by the discovery of two endoglin isoforms, L-endoglin and S-endoglin [3]. Endoglin itself does not possess kinase activity but modulates signaling by forming a heterotypic receptor complex with other TGF-β signaling receptors [4]. Mutations in the endoglin gene cause hereditary hemorrhagic telangiectasia (HHT or Osler-Weber-Rendu syndrome), a disease characterized by arteriovenous malformations [5]. Studies in endoglin heterozygous and knockout mice further confirmed the crucial role that endoglin plays in developmental angiogenesis [6,7].

The high and selective expression of endoglin on newly formed tumor blood vessels established endoglin as a predictor of poor survival in patients with various solid tumors, as reviewed in [8,9,10]. Although initially studies on endoglin were focused on its role in endothelial cells and angiogenesis, early results provided data that showed endoglin expression on macrophages [11], leukocytes, syncytiotrophoblasts in the placenta [12] and fibroblasts in culture [13]. More recent work has revealed a crucial role for endoglin expression on other non-endothelial cells, which we have recently reviewed [14]. In both endothelial and non-endothelial cells, endoglin regulates TGF-β family member signaling.

### 1.2. Endoglin Signaling

The most important and well-described ligands for endoglin are TGF-β and bone morphogenetic protein (BMP)-9. The binding of these ligands regulates endoglin expression in endothelial cells [15,16,17]. TGF-β or BMP-9 binding to endoglin in a heterotypic complex with the type-II TGF-β or BMP receptor leads to the recruitment of the activin-receptor like kinase (ALK)-1 receptor [10]. Transphosphorylation of ALK-1 by the type-II receptor leads to downstream phosphorylation of the receptor-regulated Smads-1/5/8. Subsequent complex formation with Smad4 leads to the nuclear translocation of the Smad complex and the transcription of target genes, resulting in the regulation of angiogenesis (Figure 1). Activation of the endoglin-Smad1/5/8 pathway indirectly inhibits the canonical TGF-β signaling pathway, which involves another type-I receptor, ALK-5 (Figure 1). ALK-5-dependent signaling leads to canonical TGF-β signaling and vessel maturation [18]. Ultimately, endoglin stimulates angiogenesis both directly through Smad1/5/8 signaling and indirectly by inhibition of ALK-5-mediated pathways.

In addition to ligand binding, endoglin signaling is regulated by matrix metalloproteinase (MMP)-14-dependent cleavage of the extracellular domain, which generates a soluble endoglin (sEng) variant [19]. Endoglin can also be shed through MMP-12-dependent cleavage [20]. Soluble endoglin can inhibit angiogenesis and its levels have been shown to be upregulated in pre-eclampsia [21], metabolic disorders [22] and in cancer [15], although for the latter, conflicting reports have been published. Interestingly, recent work showed that the effects of sEng (i.e., activating or inhibiting signaling) are strongly dependent on whether it circulates in the mono- or dimeric form [23]. This might provide new insights into the role and mechanisms of sEng in regulating signaling and eventually angiogenesis.

### 1.3. Targeting Endoglin for Imaging and Therapy

Given its high and selective expression on tumor blood vessels, endoglin was exploited as a target for tumor imaging. Specific tumor labeling can be accomplished by using conjugated TRC105 with PET, SPECT and NIRF imaging modalities [10,24,25]. Next to its potential for tumor imaging, endoglin was given attention as a selective tumor target for cancer therapy. Although several endoglin-targeting antibodies and peptides have been developed, the most extensive clinical development has been done using TRC105, which also was the subject of multiple preclinical studies. Below, we will summarize preclinical studies with TRC105, its clinical translation and associated biomarker studies. Finally, we will summarize the findings and discuss what we have learned to date and what questions remain.

## 2. Preclinical Studies on Endoglin Targeting

### 2.1. Development of the Anti-Endoglin Antibody Sn6j/TRC105

The first study describing the endoglin antibody Sn6j was published by Matsuno et al. in 1999 [26]. This study showed binding of Sn6j to human umbilical vein endothelial cells (HUVECs) in vitro and to tumor vasculature in severe combined immunodeficient (SCID) mice bearing human tumors. Treatment with drug conjugates of Sn6j or other endoglin antibodies showed long-lasting tumor remission of human breast cancer xenografts and the inhibition of tumor angiogenesis in vivo. Moreover, synergistic anti-tumor effects and complete regression of established MCF-7 human breast tumor nodules in mice were observed when endoglin antibodies were combined with chemotherapy [27]. The heavy and light chains of the Sn6j antibody were humanized to generate a human murine chimeric antibody that was safely administered to monkeys [28].

### 2.2. TRC105 Acts Through Immune-Dependent Mechanisms

Initial preclinical studies showed evidence that Sn6j worked through immune-dependent mechanisms. In mice bearing subcutaneous tumors, TRC105 demonstrated higher efficacy in suppressing tumor growth in immune-competent BALB/c mice compared to immunodeficient SCID mice [29]. Moreover, the combination of Sn6j with immunostimulants (CpG ODN) synergistically reduced syngeneic CT26 mouse colon tumor growth and improved the survival of BALB/c mice. These effects seemed to partly depend on CD4+, but mostly on CD8+ T-cells, since their depletion abrogated the observed anti-tumor responses [30]. Additionally, TRC105 induced apoptosis via antibody-dependent cell-mediated cytotoxicity (ADCC, Figure 2) [31]. Recent work from our groups indeed confirmed these findings, showing loss of therapeutic efficiency of TRC105 in mice lacking the Fc gamma receptors I-IV that are therefore unable to induce ADCC [32]. This study revealed that TRC105 targets a subset of intratumoral, endoglin-expressing regulatory T cells (Tregs). TRC105 appeared to eliminate this immunosuppressive population from the tumor microenvironment. These data indicate that targeting endoglin with TRC105 plays a role in regulating immune effector cells.

### 2.3. TRC105 Inhibits BMP-9-Induced Signaling

In addition to immune-dependent mechanisms, TRC105 also inhibits endoglin-dependent BMP-9 signaling (Figure 2). Strikingly, although clear anti-tumor effects in syngeneic mouse models have been shown, the binding affinity of TRC105 for mouse endoglin is significantly lower compared to human endoglin. To better study the contribution of endoglin-dependent BMP-9 signaling in preclinical models, a mouse endoglin-targeting antibody was developed, M1043 [33]. TRC105 and M1043 both very efficiently inhibit BMP-9-induced Smad1 phosphorylation in human and mouse endothelial cells, respectively, leaving TGF-β-induced, ALK-5-dependent Smad2/3 phosphorylation unaltered. Additional studies showed increased basal Smad2 phosphorylation and decreased Smad1 phosphorylation upon TRC105 treatment in HUVECs [34,35]. Although M1043 is a stronger inhibitor of BMP-9 binding to mouse endoglin than it is to human endoglin, it does appear to be less potent than TRC105 in certain in vivo colorectal cancer models [32]. The fact that the IgG subtype of M1043 (rat IgG1) does not induce ADCC in mice further strengthens the idea that ADCC is an important contributor to the therapeutic effects of TRC105.

The exact molecular mechanism of how TRC105 disturbs BMP-9 binding was more recently shown when the crystal structure of the human endoglin ectodomain and its complex with the ligand BMP-9 was published. BMP-9 interacts with a hydrophobic surface on the N-terminal orphan domain of endoglin, involving the residues mutated in HHT patients and overlapping with the TRC105 binding site [36]. As well as competing with BMP-9 binding, TRC105 induces endoglin shedding from the cell surface of endothelial cells in an MMP-14-dependent manner. This generates high levels of soluble endoglin which can act as a ligand trap and function to inhibit angiogenesis. These mechanisms of action of TRC105 might well contribute to its anti-angiogenic effects [34].

### 2.4. TRC105 and Crosstalk with the Vascular Endothelial Growth Factor (VEGF) Pathway

Acquired resistance to anti-angiogenic therapies has been reported in a multitude of studies and is mostly the result of the activation of alternative pro-angiogenic pathways in response to treatment [37,38,39]. Considerable crosstalk between the endoglin and vascular endothelial growth factor (VEGF) pathways has been reported. In colorectal cancer patient samples, it was observed that endothelial Smad1 phosphorylation was increased upon treatment with anti-VEGF therapy [35]. Furthermore, in vitro, TRC105 increased VEGF-induced ERK1/2 phosphorylation, while hardly affecting HUVEC viability [40]. In an orthotopic pancreatic cancer xenograft model, endoglin is one of three genes that underwent significant upregulation in response to VEGF blockage [41]. Combining endoglin and VEGF targeting increased efficiency in inhibiting HUVEC cord formation, fetal bone metatarsal angiogenesis assays [42] and developmental angiogenesis in zebrafish embryos [35]. In the mouse KEP1-11 breast cancer model, the combination treatment of TRC105 and VEGF targeting resulted in decreased tumor vessel density, surprisingly without significantly affecting primary tumor growth [35]. These data were confirmed in another study, where TRC105 enhanced the anti-angiogenic effects of the VEGF-A-targeting antibody bevacizumab. The combination of bevacizumab with TRC105 was shown to inhibit VEGF signaling and tip cell formation in vitro and to inhibit tumor growth, metastasis and tumor-associated angiogenesis in a murine breast cancer model [43]. In renal cancer-derived endothelial cells, the combination of TRC105 and the multi-tyrosine kinase inhibitor sunitinib induced phosphorylation of Smad2/3 to promote endothelial cell death. Moreover, TRC105 enhanced the inhibitory effect of sunitinib on VEGF signaling and reduced VEGFR2-Akt-Creb activation, suggesting a molecular cooperation between the two drugs [44]. Together, these data show significant crosstalk between endoglin and the VEGF pathway and provided the basis for combination studies in patients, as discussed later in this review.

### 2.5. TRC105 and Anti-Tumor Effects in Preclinical Cancer Models

TRC105 has been tested in a wide variety of preclinical cancer models, most of them reporting the inhibition of tumor growth and sometimes even complete tumor regression, although resistant cancer types have also been reported, like the KPC3 pancreatic cancer model (Schoonderwoerd et al., submitted). TRC105 was shown to inhibit the growth of syngeneic CT26 and MC38 mouse colon cancer and MCF7 and 4T1 breast cancer models [45].

TRC105 was additionally tested in multiple other cancer mouse models. Dourado and colleagues [46] showed that TRC105 treatment prevented human acute myeloid leukemia (AML) blast engraftment in irradiated NOD SCID gamma (NSG) mice, and, when administered at disease onset, inhibited AML progression in vivo. In acute B-lymphoblastic leukemia (B-ALL), TRC105 alone was ineffective in inducing a long-term response, which might be explained by the observation that B-ALL blasts produce high levels of sEng which could result in a sink effect for TRC105, preventing effective target binding on the tumor cells. However, in both B-ALL and AML, TRC105 synergized with chemotherapy to inhibit leukemia development or disease progression, respectively.

In mouse models for renal cancer, TRC105 impaired the ability of tumor endothelial cells (TECs) and cancer stem cell-derived TECs to organize into tubular structures, whereas it did not limit proliferation or survival. The combination of TRC105 with sunitinib synergistically reduced tube formation, proliferation and survival of CSC-TECs and tumor-derived TECs to a similar extent [44].

In mice bearing ovarian cancer cells, TRC105 treatment resulted in decreased metastatic spread of high-grade serous ovarian cancer, reduced ascites and delayed growth and spread of abdominal tumor cells, thus extending overall survival in vivo [47].

The anti-tumor effects of TRC105 were initially mostly assigned to targeting of the tumor blood vessels, restricting oxygen and nutrient availability, thereby reducing tumor growth. However, more recent studies revealed additional targets of TRC105 therapy apart from endothelial cells.

In the KEP1-11 breast cancer model, treatment with TRC105 reduced the presence of cancer-associated fibroblasts (CAFs) in primary tumors and reduced metastatic spread in an adjuvant setting [35]. In prostate cancer models, TRC105 treatment significantly potentiated the anti-tumor effects of radiation therapy, when compared to irradiation alone [48]. Additional research showed that the combination of androgen deprivation therapy (ADT) and TRC105 reduced castration-resistant prostate cancer progression through interruption of the communication between endoglin-expressing CAFs and prostate cancer cells [49].

Moreover, in colorectal cancer (CRC), endoglin-expressing CAFs seem to contribute to metastatic potential and the formation of liver metastasis in an experimental model of CRC-derived liver metastases in nude mice. These pro-metastatic effects of CAF-specific endoglin could be inhibited by TRC105, indicating CAFs as an additional TRC105 target, in addition to the angiogenic endothelial cells [50].

Finally, two studies reported on the combined effects of TRC105 and immunotherapy. In a neuroblastoma study in immunodeficient NSG mice, it was shown that immunotherapy with the anti-GD2 antibody dinutuximab combined with activated NK cells is suppressed by endoglin-positive cells in the tumor microenvironment. This suppression could be overcome by targeting endoglin on the human mesenchymal stem cells (MSCs) and murine endothelial cells and macrophages using TRC105 and M1043, respectively, and resulted in improved survival [51]. Furthermore, the combination of TRC105 with the checkpoint inhibitor PD1 was shown to be effective in inhibiting tumor initiation in a chemically induced model for colitis-associated CRC and subcutaneous and orthotopic syngeneic MC38 and CT26 CRC models. Importantly, sustained anti-tumor and memory responses were reported [32].

Taken together, these studies show that TRC105 treatment, alone or as a combination therapy, seems effective in inhibiting tumor growth and preventing metastatic spread in preclinical cancer models. The effects are partly from targeting tumor angiogenesis, but also extend to other cell types, including CAFs, Tregs and other cells suppressing therapeutic responses like myeloid-derived suppressor cells. The expression of endoglin on a variety of cells increases the potential for TRC105 to be used as a cancer therapy and might even extend beyond the field of oncology, looking, for example, at fibrotic diseases.

## 3. TRC105 Clinical Trials

### 3.1. Biomarker Findings in Clinical Trials

Consistent with what has been noted for the entire class of anti-angiogenic agents, not all patients respond to TRC105. To enrich patients most likely to benefit, as well as to investigate the molecular impact of the drug, blood-based biomarkers were analyzed using multiplex ELISA to quantitatively assess circulating proteins related to angiogenesis, inflammation and immunity. This panel, termed the angiome [52], is a protein multiplex array that was consistently applied to several TRC105 clinical trials in order to explore potential prognostic, predictive and pharmacodynamic biomarkers. A list of all the clinical trials testing TRC105 is shown in Table 1. Below, we discuss some of the trials and their biomarker findings.

### 3.2. TRC105 First-in-Human Clinical Trial

Based on the anti-tumor potency in various preclinical models, as described above, TRC105 was first tested in a phase I trial (NCT00582985) in 2008 that enrolled 50 cancer patients to assess safety and efficacy [53]. The recommended phase II dose was established to be 10 mg/kg weekly or 15 mg/kg every two weeks. In this trial, stable disease or better was achieved in 21 of 45 evaluable patients. Two patients demonstrated prolonged responses. One prostate cancer patient showed a complete prostate-specific antigen (PSA) response with bone scan normalization for more than 5 years, and a heavily pretreated uterine carcinosarcoma patient was progression free for 1.5 years.

The angiome was tested in plasma samples from 32 of 50 patients (64%) at baseline, during treatment and the end of the study [54]. Several VEGF family members, such as PlGF, VEGF and VEGF-D decreased significantly following TRC105 treatment, suggesting an overall suppression of angiogenesis. Interestingly, this effect is distinct from bevacizumab, which increases the level of these markers [54]. Soluble endoglin (sEng) also increased in a dose-dependent manner. sEng induction in response to TRC105 could be recaptured in endothelial cell culture [34,40] and represents the strongest pharmacodynamic effect elicited by TRC105. Taken together, the first-in-human study for TRC105 showed that treatment was safe, with indications of clinical activity and formed the basis for further clinical development.

### 3.3. Crosstalk with the VEGF Pathway: TRC105 Plus VEGF Inhibitors in Phase Ib Trials

Given the considerable crosstalk between endoglin and VEGF pathways, as described above, targeting both angiogenic pathways simultaneously was expected to exert more potent anti-angiogenic and therefore anti-tumor effects. Between 2011 and 2013, a phase Ib, dose-finding study of TRC105 in combination with the VEGF antibody bevacizumab was conducted (NCT01332721) [55]. Both drugs were well tolerated at their recommended single agent doses (10 mg/kg/week for TRC105 and 10 mg/kg/2 weeks for bevacizumab).

Impressively, combining TRC105 with bevacizumab demonstrated durable activity in a VEGF inhibitor refractory patient population [55]. Drug activity was observed in 18 cancer patients (47%), including two partial responses (PR) and 16 patients with prolonged stable disease (SD). Approximately 25% of patients experienced a 10–25% tumor volume reduction and remained progression free for periods longer than experienced on prior anti-VEGF therapy. CT scans also revealed favorable tumor morphology change, such as decreased tumor density [56]. The angiome biomarker panel was assessed across 37 patients (97%) at baseline, after bevacizumab lead-in monotherapy, after co-administration of both drugs for 3 weeks and at the end of treatment [57]. The modulation of certain pharmacodynamic biomarkers, such as bevacizumab-induced PlGF increases and TRC105-induced sEng increases, were observed and not affected by the presence of the other drug. Additionally, novel biomarker modulations were identified when the two drugs were used in combination. For example, bevacizumab treatment increased VEGF-D levels [58], while TRC105 monotherapy decreased VEGF-D [54]. In response to the combination of drugs, VEGF-D levels did not statistically differ, indicating compensatory VEGF-D signaling was inhibited, possibly accounting for delayed drug resistance [57].

### 3.4. TRC105 Plus VEGF Inhibitors in a Series of Phase Ib Trials

The dense expression of endoglin on blood vessels in highly vascularized tumors led to the study of TRC105 in glioblastoma [59], renal cell carcinoma (RCC) [60], hepatocellular carcinoma [61] and breast cancer [62]. In addition, endoglin expression on RCC cells [63] and certain sarcoma subtypes [64,65] made these tumor types highly relevant for endoglin targeting.

TRC105 was therefore studied in combination with VEGF inhibitors in glioblastoma [66], sarcoma [67,68], RCC [69,70], breast cancer [71], metastatic castration-resistant prostate cancer [48,72], hepatocellular carcinoma [73,74] and urothelial carcinoma [75]. The angiome was tested in four of these phase 1b trials, with three trials combining TRC105 with VEGF inhibitors and one trial testing TRC105 in combination with chemotherapy alone.

#### 3.4.1. Glioblastoma (NCT01648348)

In this study, TRC105 at a dose of 10 mg/kg/week was given to 22 recurrent or progressive GBM patients with or without bevacizumab. For patients receiving dual therapy, TRC105 began after one week of bevacizumab lead-in monotherapy. Median progression-free survival (PFS) for the five patients receiving only TRC105 was 1.38 months, while median PFS for the 14 patients receiving both TRC105 and bevacizumab was 1.81 months (95% CI: 1.25—2.07). The angiome analysis revealed a marked increase of sEng. However, an increase in PlGF and VEGF, as well as a decrease in VEGF-R2, typical changes following bevacizumab treatment, were not observed, suggesting biomarker modulation may be different in this disease type.

#### 3.4.2. Sarcoma (NCT01975519)

TRC105 (8 or 10 mg/kg/week) combined with pazopanib was given to 19 patients with advanced soft tissue sarcoma after one cycle of pazopanib lead-in monotherapy. Six patients responded according to Choi criteria, reaching a response rate of 32%. In the six responders, baseline levels of ICAM-1 and TSP-2 were significantly lower. Typical biomarker modulations in response to VEGF inhibitors, such as upregulation of PlGF, VEGF and VEGF-D, as well as downregulation of VEGF-R2 and TSP-2, were noted as significant on-treatment changes.

#### 3.4.3. Renal Cell Carcinoma (NCT01806064)

TRC105 (8 or 10 mg/kg/week) together with axitinib was administrated to 18 patients with advanced or metastatic renal cell carcinoma (mRCC). Five patients exhibited a more than 30% reduction in tumor size, demonstrating an ORR of 28%. In the five responders, the baseline level of osteopontin (OPN) was lower, while TGFbeta (β)-R3 was higher [56]. Consistently, we noted an increase in sEng as a TRC105-specific effect and an increase in PlGF, VEGF and VEGF-D, as well as a decrease in TSP-2 and VEGF-R2, as a VEGF inhibitor-specific effect.

Across the three trials, VEGF monotherapy (bevacizumab for 1 week in GBM, pazopanib for 1 cycle in sarcoma) had little impact on sEng levels. In contrast, sEng increased in all three patient populations following TRC15 treatment, validating sEng as an on-target pharmacodynamic marker (Figure 3).

Beyond representing a pure pharmacodynamic marker, sEng has important biological functions, such as attenuating the proliferation, migration and tube formation of human umbilical vein endothelial cells [76]. In addition, sEng has been shown to reduce inflammation in mouse models [77]. Alternatively, it has been reported that increased circulating sEng may preferentially direct BMP-9 signaling via cell surface endoglin at the endothelium rather than being an inhibitory ligand trap [23]. The exact function of sEng in patients receiving TRC105 remain to be clarified.

No other consistent, TRC105-dependent modulations were noted for other markers within the endoglin signaling pathway, including upstream ligands (BMP-9, TGF-β1), receptors (TGFβ-R3) and downstream effectors (PDGF-AA, PDGF-BB).

Specific biomarker changes could be identified in response to monotherapy with VEGF inhibitors. Increases in PlGF, VEGF and VEGF-D, as well as decreases in TSP-2 and VEGF-R2, are all noted pharmacodynamic changes in response to VEGF inhibitors, consistent with the literature [58,78,79]. In contrast, VEGF-R3, a receptor playing crucial roles in lymphangiogenesis, significantly decreased in response to both VEGF inhibitors and TRC105, suggesting VEGF-R3 modulation is not specific to VEGF inhibitors and may represent a broader effect of anti-angiogenic therapies.

#### 3.4.4. TRC105 and Capecitabine in Breast Cancer Patients (NCT01326481)

In this trial, TRC105 (7.5 or 10 mg/kg/week) was combined with capecitabine and administrated to 19 progressive or recurrent metastatic HER2-negative breast cancer patients. There were four responders in this study. Angiome analyses revealed that baseline sEng levels were noted to be higher in the responders compared to the non-responders; however, this difference did not reach statistical significance (*p* = 0.078). In addition to a well-defined upregulation of sEng levels, the endoglin ligand BMP-9 was significantly increased in the circulation of most patients (71%), potentially reflecting a compensatory mechanism. No significant changes in PlGF, VEGF, VEGF-D, TSP-2 or VEGF-R2, were detected in this trial, offering further evidence that changes in these five markers are specific to the combination of TRC105 plus VEGF inhibitors.

### 3.5. TRC105 Biomarkers in Randomized Trials

Encouraged by these promising phase 1b studies, randomized phase 2/3 trials were next performed in mRCC and angiosarcoma. In the TRAXAR trial, 150 mRCC patients were randomized in a 1:1 ratio to standard dose axitinib or TRC105 combined with axitinib (NCT01806064) [80]. In the TAPPAS trial, 128 angiosarcoma patients were randomized to pazopanib or TRC105 and pazopanib (NCT02979899) [81]. In both studies, no improvement in PFS was observed by the addition of TRC105 to VEGF inhibitors, leading to termination of the further development of TRC105 [80,81].

While several circulating biomarkers were shown to be prognostic for outcome in earlier studies, the identification of a biomarker that could identify patients most likely to respond to TRC105 remains an unmet need. The randomized design of the two studies mentioned above enables the discovery of predictive biomarkers for TRC105. While the angiome was not tested in samples collected from patients in the TAPPAS trial, it has been assessed in patients from the TRAXAR trial. Interestingly, VEGF was identified as a potential predictive marker [80]. Other analyses in randomized trials of TRC105 also indicated potential predictors of efficacy. In a separate phase 2b randomized trial, mRCC patients were treated with bevacizumab alone or bevacizumab + TRC105. PFS was not improved by the addition of TRC105. The authors reported that lower TGF-β levels (<10.6 ng/mL) are associated with better PFS at the 12- or 24-week landmarks [70]. It remains possible that a sub-population of patients, guided by proper biomarkers, would benefit from the addition of TRC105 to VEGF inhibitors.

### 3.6. Biomarker Conclusions

By now, the angiome has been assessed in seven phase 1-2 trials featuring TRC105 monotherapy or a combination with VEGF inhibitors. In short, TRC105 induces distinct biomarker modulations from VEGF inhibitors. sEng has been identified as a strong pharmacodynamic marker, exhibiting a direct drug effect of TRC105. sEng is an important marker in other diseases, such as pre-eclampsia and Osler-Weber-Rendu syndrome [82]. The identification of VEGF as a potential predictive marker in the randomized mRCC trial emphasizes the importance of patient pre-selection to achieve precision medicine. The lesson learned from the angiome analysis across all TRC105 trials will be appliable to novel anti-angiogenic drugs.

## 4. Discussion

### 4.1. Lessons Learned, Question to Be Answered

During the last 20 years, many studies on targeting endoglin, as either monotherapy or combined with other (anti-angiogenic) therapies, have been performed. Although initial encouraging results were reported, the pivotal trials for TRC105 did not demonstrate clinical benefit to warrant further clinical development. Despite this disappointing result, many of these studies have revealed valuable knowledge on endoglin biology, endoglin expression on target cells and crosstalk with other pro-angiogenic pathways.

Endoglin-targeting therapy does not seem to fit the “classical” anti-angiogenic therapies. Although clear crosstalk between the endoglin and VEGF pathways has been shown, combined TRC105/anti-VEGF therapy appeared to be effective in VEGF therapy refractory patients and in preclinical models. These observations might be explained by the binding of TRC105 to additional target cells. Endoglin expression has been reported on tumor-infiltrating Tregs, macrophages, CAFs and cancer (stem) cells in human and mouse samples. This could contribute to the efficiency of TRC105, since targeting those cells might enhance anti-tumor responses. Intriguingly, a decrease in Tregs was also observed in patients dosed with TRC105 [53], showing additional evidence for the targeting of endoglin-expressing Tregs by TRC105. Furthermore, targeting endoglin on fibroblasts might also extend beyond oncology. Endoglin expression has been shown on activated fibroblasts in cardiac fibrosis, where targeting endoglin with TRC105 in preclinical models reduced cardiac fibrosis and improved outcomes [83,84]. Since several studies have investigated the role of endoglin and its ligand BMP-9 in the progression of liver fibrosis, this might be an interesting field of study.

The regulation of angiogenesis by BMP-9 and TGF-β has been a subject of debate for quite some years since their effects seem very concentration and receptor dependent. Despite this fact, the inhibition of endoglin-dependent BMP-9 signaling via TRC105, sEng or by using the BMP-9-binding ligand trap ALK1-Fc [85,86] reduces angiogenesis. However, the inhibition of BMP-9 signaling only seems partly responsible for the therapeutic effects. The immune-dependent effects and induction of ADCC seem to be of crucial importance for the anti-tumor effects of TRC105. This was further emphasized by preclinical studies showing that immunomodulatory therapies increase TRC105 efficiency. Unfortunately, a clinical trial with TRC105 and the PD1 inhibitor nivolumab (NCT03181308) was terminated prematurely. Therefore, it remains unknown if this also holds true for cancer patients.

### 4.2. Looking Forward

All studies have increased our knowledge on endoglin biology, its expression and regulation, crosstalk with other pathways and appropriateness as a therapeutic target in, and beyond, oncology. Obviously, it is disappointing that despite very promising preclinical studies, the clinical development is only ongoing for acute macular degeneration (AMD). One could argue that if TRC105 was tested in the right setting or in earlier stages of disease, it could prove more effective. Alternatively, immunologically “hot” tumors might also be more responsive given the strong involvement of the immune system and the increased efficiency when combined with checkpoint inhibition in mouse models. One questions that still remains is how on a functional level endoglin exerts its pro-tumorigenic effects. This might be the regulation of immune cell infiltration or affecting the migratory properties of cells. Current in vitro models poorly recapitulate these processes, while studies in mice are hampered by the differences in mechanisms of action and binding of TRC105. Many more years of research should shed light on this and reveal if a novel approach for endoglin targeting would be valuable.

## Figures and Tables

**Figure 1 ijms-22-00147-f001:**
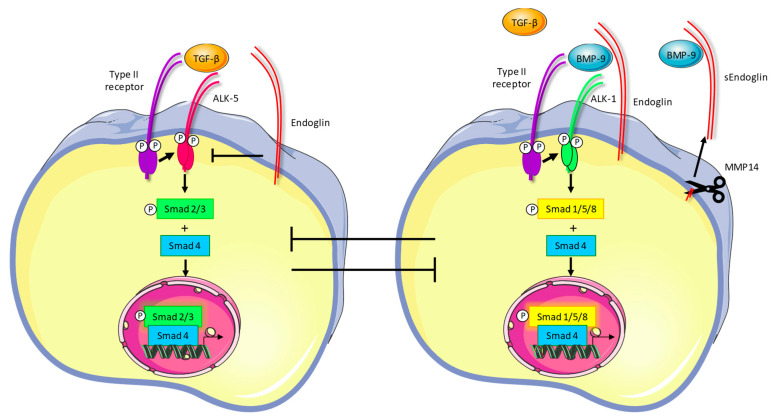
Endoglin signaling. When TGF-β binds to its type-II receptor in the absence of endoglin, the type-I receptor ALK-5 is recruited and transphosphorylated. This subsequently leads to phosphorylation of Smad2/3, which then forms a complex with Smad4, translocates to the nucleus and regulates target genes involved in vessel maturation. The presence of endoglin on the cell membrane inhibits this signaling pathway (left panel). In the presence of endoglin, binding of BMP-9 or TGF-β to the type-II receptor results in recruitment and transphosphorylation of ALK-1, which phosphorylates Smad1/5/8, resulting in regulation of angiogenic target genes (right panel). Endoglin can be cleaved from the cell membrane by MMP-14, creating a soluble form of the receptor which may function as a ligand trap.

**Figure 2 ijms-22-00147-f002:**
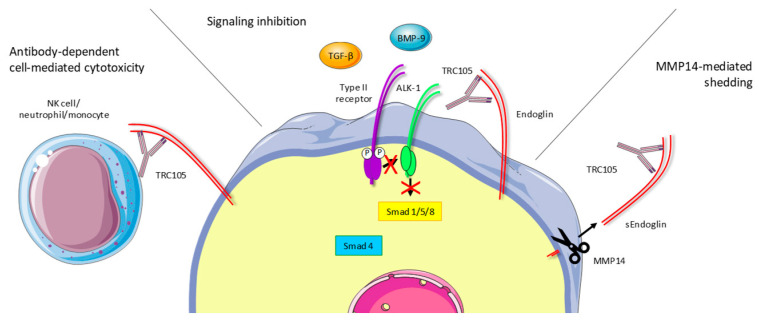
Proposed mechanisms of action of TRC105. TRC105 exerts its actions through different mechanisms. Firstly, binding to endoglin induces antibody-dependent cell-mediated cytotoxicity (ADCC) mediated by NK cells, neutrophils and monocytes, resulting in target cell killing. Secondly, TRC105 prevents BMP-9 binding to endoglin, thereby inhibiting receptor complex formation and endoglin-mediated downstream signaling. Lastly, MMP-14-mediated endoglin shedding is induced upon TRC105 binding, generating soluble endoglin (sEng) which can function as a ligand trap and thus reduces endoglin signaling.

**Figure 3 ijms-22-00147-f003:**
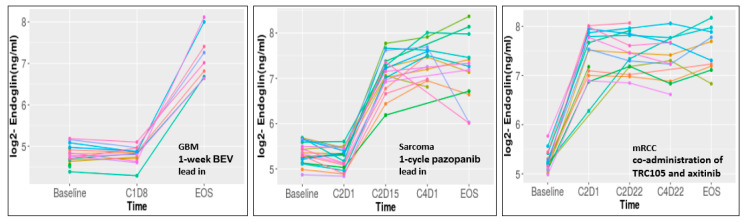
TRC105 increases sEng levels in circulation of patients. Each individual patient is represented by a colored line.

**Table 1 ijms-22-00147-t001:** TRC105 clinical trials.

Trial Description	NCT Number	Trial Phase	Patients Enrolled	Primary Outcome
Open label phase 1 dose-finding study of TRC105 in patients with solid cancer	NCT00582985	1	51	Safety
Preoperative combination of letrozole, everolimus and TRC105 in postmenopausal hormone receptor-positive and Her2-negative breast cancer	NCT02520063	1/2	14	MTD
Open label dose-finding study of TRC105 plus capecitabine for metastatic breast cancer (TRC105)	NCT01326481	1/2	19	MTD
A phase I/II study of TRC105 in metastatic castration-resistant prostate cancer (CRPC)	NCT01090765	1/2	21	MTD
Sorafenib and TRC105 in hepatocellular cancer	NCT01306058	1/2	27	MTD and TTP
Open label continuation study of TRC105 for patients who have completed a prior TRC105 trial	NCT02354612	1/2	50	Long-term TRC105 response
Bevacizumab with or without anti-endoglin monoclonal antibody TRC105 in treating patients with recurrent glioblastoma multiforme	NCT01648348	1/2	116	MTD, safety and PFS
Study of carotuximab (TRC105) plus nivolumab in patients with metastatic NSCLC	NCT03181308	1b	11	Safety
Study of TRC105 combined with standard-dose bevacizumab for advanced solid tumors for which bevacizumab is indicated	NCT01332721	1b	38	MTD
A study of TRC105 in combination with paclitaxel/carboplatin and bevacizumab in non-squamous cell lung cancer	NCT02429843	1b	16	Change in medical management
A phase 1B dose-escalation and phase 2a study of carotuximab (TRC105) in combination with pazopanib in patients with advanced soft tissue sarcoma	NCT01975519	1b, 2	30, 89	Safety, PFS, ORR
Trial of TRC105 and sorafenib in patients with hepatocellular carcinoma (HCC)	NCT02560779	1b/2	27	Safety and ORR
TRC105 for recurrent glioblastoma	NCT01778530	2	2	Radiographic response rate
TRC105 combined with standard-dose bevacizumab for two patients with metastatic and refractory choriocarcinoma	NCT02396511	2	2	PFS and ORR
Study of TRC105 and bevacizumab in patients with refractory gestational trophoblastic neoplasia (GTN)	NCT02664961	2	3	ORR
TRC105 for liver cancer that has not responded to sorafenib	NCT01375569	2	11	TTP
Study of TRC105 with abiraterone and with enzalutamide in prostate cancer patients progressing on therapy	NCT03418324	2	11	Disease stabilization or improvement at 2 months
TRC105 in adults with advanced/metastatic urothelial carcinoma	NCT01328574	2	13	PFS
Study of TRC105 + paclitaxel/carboplatin and bevacizumab in patients with NSCLC	NCT03780010	2	15	Safety
A phase 2 evaluation of TRC105 in combination with bevacizumab in patients with glioblastoma (105GM201)	NCT01564914	2	22	OS
Evaluation of TRC105 in the treatment of recurrent ovarian, fallopian tube or primary peritoneal carcinoma	NCT01381861	2	23	PFS, ORR, safety
Bevacizumab with or without TRC105 in treating patients with metastatic kidney cancer	NCT01727089	2	59	PFS
Randomized phase 2 trial of axitinib and TRC105 versus axitinib alone in patients renal cell carcinoma	NCT01806064	2	150	Safety and PFS
Trial of TRC105 and pazopanib versus pazopanib alone in patients with advanced angiosarcoma (TAPPAS)	NCT02979899	3	128	PFS

MTD: maximal tolerated dose; TTP: time to tumor progression; PFS: progression free survival; OS: overall survival; ORR: objective or overall response rate.

## Data Availability

All data previously exist in the literature. No new data was generated for this review.

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
