# Peer review of "Endoglin Targeting: Lessons Learned and Questions That Remain"

_ijms, 2020, doi:10.3390/ijms22010147_

Round 1
Reviewer 1 Report
Authors present here very nice comprehensive review focusing on possible influence of endoglin with respect to the cancer. This is nice review updating the recent work in endoglin field.
Minor comments and suggestions that could improve the manuscript:
- I suggest adding short figure legend describing the scheme in figure 1 despite the description is in the text.
- The same is valid for figure 2 however I am not sure if the text related to figure 2 below is part of the figure or part of the text. I suggest being part of the text of the figure because figure should be self-explanatory.
- Authors claim that “TRC105 and M1043 both very efficiently inhibit BMP-9-induced Smad1 phosphorylation in human- and mouse endothelial cells, respectively, leaving TGF-β-induced, ALK-5-dependent Smad2/3 phosphorylation unaltered” Are authors sure there are no studies even not related to angiogenesis and cancer TRC105 does not affect Smad 2/3 at all?
- Authors should comment/speculate whether increased sEng levels after TRC105 treatment in clinical trials are high enough and could affect endoglin signaling in tumors or they are just biomarker of the treatment.
Author Response
Dear reviewer, thank you for your careful checking. We have addressed your concerns, please see below.
- Detailed figure legends have been added to both Figure 1 and 2, as shown on page 2, line 70; page 4, line 33.
- As for the effect of TRC105 on Smad2/3 phosphorylation, it is much less potent than its inhibitory impact on Smad1/5/8. We cited several papers investigating this issue, including Liu et al (ref 40).
- After TRC105 administration, sEng markedly increases, yet its function remains controversial. sEng may act as a ligand trap, but it may also help bringing BMP9 to membrane bound Endoglin, exerting either inhibitory or promoting effect. We added a paragraph to clarify this point on page 9, line 320-325.
Hope we answered your questions satisfactorily. Thanks again.
Reviewer 2 Report
The article presented by Liu Y. et al. is an interesting overview on Endoglin targeting, the TGF-β coreceptor, which has an important role in particular in cancer angiogenesis.
The authors showed and discussed the effects of anti-endoglin neutralizing antibody TRC105 (Carotuximab) tested in a high variety of preclinical cancer models as well as phase I-III clinical studies in cancer patients.
Although initial encouraging results were reported, the crucial trials for TRC105 did not demonstrate clinical benefits. Despite this inconclusive result, this review in my opinion brings new light on endoglin biology as well as on crosstalk with other pro-angiogenic pathways in the tumor microenvironment.
The article is well presented and well discussed. In my opinion, in the present form the article is ready to be accepted.
Author Response
Dear reviewer,
Thank you for your kind words and encouragement. We really appreciate it.
Best regards
Yingmiao Liu, Ph.D.
Duke University Medical Center
Durham, NC
USA